# An Acid-Free Alternative to Pectin Production from the Cell Walls of Olive Oil Waste and Different Fruits Using Choline Chloride

**DOI:** 10.3390/foods12224166

**Published:** 2023-11-17

**Authors:** Alejandra Bermúdez-Oria, María Luisa Castejón, África Fernández-Prior, Guillermo Rodríguez-Gutiérrez, Juan Fernández-Bolaños

**Affiliations:** Department of Food Phytochemistry, Instituto de la Grasa (Spanish National Research Council, CSIC), Pablo Olavide University, Building 46, Ctra de Utrera km 1, 41013 Seville, Spain; aleberori@ig.csic.es (A.B.-O.); mcastejon1@us.es (M.L.C.); mafprior@ig.csic.es (Á.F.-P.); guirogu@ig.csic.es (G.R.-G.)

**Keywords:** pectin, choline chloride, alcohol-insoluble residue (AIR), cell wall, NMR and IR spectroscopy, deep eutectic solvents (DES)

## Abstract

The pectin from the cell walls of olive waste (alperujo) and apple, orange and strawberry fruits was extracted using choline chloride (ChCl) and the yield and chemical and structural compositions were compared to pectin extracted using citric acid (CA) and ammonium oxalate/oxalic acid (AOOA). According to the results, the alperujo pectin extracted using ChCl from alcohol-insoluble residue (AIR) showed a higher yield (2.20–2.88% on the basis of dry weight of AIR) than using CA (0.65–1.22%) but lower than using AOOA (3.92–5.42%). For fruit pectin, the highest yield was obtained using CA (8.81–16%), followed by AOOA (5.4–6.63%), although for apple pectin, ChCl gave a similar yield (5.36%) to AOOA. The uronic acid contents in all ChCl pectins (45.9–70.6% dry basis AIR) were higher or similar to that of the other extracting agents (30.6–65.2%), although a lower level of neutral sugar side chains was detected, with a lower degree of branching and degree of methylation. The NMR and FT-IR spectroscopy of the pectin isolated using ChCl confirmed its slightly different structural composition with respect to CA and AOOA pectin. Therefore, depending on the source material and functionality, pectin isolated using ChCl could be an acid-free alternative to pectin production.

## 1. Introduction

Pectin is a heteropolysaccharide located in the middle lamella complex of the primary cell walls of fruits and vegetables [1]. It consists of galacturonic-acid-based polysaccharides, homogalacturonan (HG), rhamnogalacturonan I (RG-I), rhamnogalacturonan II and xylogalacturonan are the main types commonly isolated and purified [2]. HG is a linear structure of α-1,4-D-galacturonic acid units with poor solubility which can be partially methyl- and acetyl-esterified [3]. RG-I contains a backbone of alternating rhamnose and galacturonic acid residues, which can be branched by several neutral polymers such as arabinans, galactans, and arabinogalactans [4]. Pectin is widely used for its technological properties in the food industry as a gelling and stabilizer agent, thickener, and emulsifier, but it also presents functional properties associated with reducing the blood cholesterol level, improving immunity, and showing anti-tumor activity [5]. However, the chemical and macromolecular characteristics of pectin and its functional properties and potential uses depend mainly on the structural properties of the source of the native pectin plant and on the impact of the extraction method [6]. High-methoxyl (HM) pectin is widely used as a texturing agent for the production of jams and jellies, while low-methoxyl (LM) pectin can be used as a fat replacer in spreads, ice cream, emulsified meat, or low-calorie products, among others [7].

The interaction between pectin chains and pectin with other cell wall polysaccharides (cellulose and hemicelluloses) and proteins depends on various crosslinks. These include ionic bridges between HG chains via divalent ions (Ca++), covalent bonds between uronic acid and the hydroxyl group of the neighboring polysaccharide, hydrophobic interactions between methoxyl groups, and hydrogen bonds between the carboxyl and alcohol groups [8]. Therefore, pectin extraction is a complex process in which the molecular interaction conditions the hydrolysis and solubilization of different types of pectin from the cell wall, and is influenced by different factors such as solvent, temperature, pH, time, and material/solvent ratio [9]. Depending on the solvent used, hot water, chelating agents, dilute mineral acids (HCl, H_2_SO_4_, HNO_3_), or organic acid (citric acid), the extractable pectin has specific structural domains which imply specific properties both from a technological and functional point of view [10]. Conventional extraction methods using dilute mineral acids, currently replaced by organic acids with a high acidity and high temperature, could influence the purity and quality of the extracted pectin. The use of chelators such as citric acid and oxalic/oxalate under acidic conditions has been considered a good pectin extracting agent with respect to the yield and physicochemical properties of the pectin [7,11,12]. However, the industry currently avoids the use of organic acid in pectin extraction because it is a more expensive process than mineral acid since organic acid requires a higher amount of acid to reach a certain pH. The process must be sustainable and economical [13]. For this reason, the search for an acid-free extraction process, in addition to hot water, with good yields and good physicochemical and bioactive properties is an ambitious challenge.

Emerging technologies such as ultrasound, microwaves, supercritical water extraction, or enzymatic treatments have been developed to extract pectin, although their scale-up is problematic due to high investment costs [10]. In addition, new solvents with low-cost and environmentally friendly properties, such as deep eutectic solvents (DES), have been tested as potential solvents for pectin extraction. These solvents are eutectic mixtures composed of at least two components in a certain molar ratio, a hydrogen bond acceptor (HBA), most predominantly quaternary ammonium salt, and hydrogen bond donors (HBD) from a wide range of compounds such as sugars, organic acids, and/or alcohols. The hydrogen bonds between them are responsible for the formation of DES. These DES provide an additional molecular interaction with the wall material and deconstruct it, helping to extract pectin [14,15]. Although DES are cheap and relatively easy to prepare, they require mixing of the compounds in order to reach the eutectic point. It could be thought that these DES are safe and their toxicity and biodegradability is low. However, there is some controversy surrounding the greater negative effects depending on the composition, viscosity, and concentration in the eutectic mixture than in the individual components [16]. In addition, the high viscosity of DES and the fact that it is not easy to separate them from the final product can make their use difficult [9,10]. In the case of carboxylic acid–choline chloride (ChCl)-based DES, the most commonly used for pectin extraction, their applicability is compromised because they are degraded by the esterification reaction between the carboxylic acid and the alcohol moiety of the ChCl [17]. Also, some authors suggest the possibility of using an aqueous solution of choline chloride to fractionate the lignocellulosic biomass, extracting lignin and improving the hemicellulose-derived stream, instead of pre-treatments using DES comprising ChCl and carboxylic acids [18,19].

In this study, the possibility of using quaternary ammonium salt ChCl as an inexpensive, available, non-toxic, low-viscosity agent with a good hydrogen bonding capacity in the isolation of pectin is examined. We investigated the use of ChCl alone, and not in the eutectic mixture, as well as whether any interaction between the ChCl and certain components of the complex cell wall via hydrogen bonds, as is the case with eutectic solvents, could favor pectin solubilization.

In preliminary assays, the application of ChCl to olive waste (alperujo) revealed the extraction of pectic polysaccharide, which presented bound phenols [20]. Since the polyphenols attached to pectin can influence, especially in the case of alperujo, the purity and quality of the extracted pectin, we investigated in this work the use of ChCl for the isolation of pectin from the cell wall without interference. We tested the extraction of pectin from the cell walls as alcohol-insoluble residue (AIR), which reduces the content of extractives, removing part of the phenols, pigments, and minerals, among others. To achieve this objective, taking into account significant variation among olive waste, three different olive waste samples (ALP-1, ALP-2, ALP-3) were used. In addition, in order to compare this behavior of ChCl in the extraction of pectin from alperujo with other plant sources, the ChCl pectin extraction from three different types of fruit (apples, oranges, and strawberries) was also assessed. In all cases, the yield and the chemical and structural compositions of the ChCl pectin extracted, as a measure of the quality of the pectin, was compared to two commonly used pectin extraction agents such as citric acid (CA) and ammonium oxalate/oxalic acid (AOOA) [7,11,21]. Therefore, our aim was to confirm these preliminary results and evaluate whether ChCl is a potential extracting agent of cell wall pectin which improves the selectivity and purity of the extracts.

## 2. Materials and Methods

### 2.1. Raw Material

Two different fresh samples of olive waste (alperujo) (a semi-solid residue composed of olive peel, pulp, seeds, and ground stones) were collected, at the beginning (ALP-1) and in the middle (ALP-2) of the harvesting season, directly from the end of the horizontal centrifuge of the two-stage extraction system from an experimental virgin oil mill of the Instituto de la Grasa (CSIC) (Seville, Spain). The third sample of alperujo (ALP-3) was supplied by a pomace oil extraction factory (Marchena, Seville, Spain) after a certain period (unknown) of storing the olive paste. In all three cases, the samples were stored at 4 °C until used.

The oranges, apples, and strawberries were purchased from a local market (Seville, Spain) and were used immediately.

### 2.2. Chemicals

The choline chloride was purchased from Thermo Scientific (Kandel, Germany). The citric acid was obtained from Merck (Darmstadt, Germany). The oxalic acid was purchased from Acros Organics (Morris Plains, NJ, USA) and the ammonium oxalate, galacturonic acid, m-hydroxydiphenyl, trifluoroacetic acid, rhamnose, fucose, arabinose, xylose, mannose, galactose, glucose, dialysis bags of cellulose with a molecular cut-off of 12,000 Da, and alcohol oxidase were purchased from Sigma-Aldrich-Fluka (St. Louis, MO, USA).

### 2.3. Preparation of Cell Wall Material

Alcohol-insoluble residue (AIR) from the fruits and the three samples of alperujo was obtained using the cold alcohol-insoluble residue method of Renard (2005) [22]. Briefly, the apples and orange were peeled and the calix or green hull was removed from the strawberries. All the fruits were cut into 2–3 pieces and the alperujo samples (1.0–1.5 kg) were suspended in 2–4 L of 96 mL/100 mL of ethanol solution (volume fraction), depending on the moisture of the sample, at room temperature and directly ground in a domestic blender for 60 s and filtered through nylon cloth. The resulting paste was suspended in 500 mL of 70 mL/100 mL ethanol solution and ground for bursts of 15 s and filtered and washed with additional ethanol (70 mL/100 mL) on a nylon cloth. This process was repeated until the filtrate had no color. This solid was dried using solvent exchange with 96% ethanol three times, and pure acetone three times, and kept overnight in an oven at 40 °C.

### 2.4. Extraction of Pectin from the Cell Wall

The extraction methods were carried out using choline chloride (ChCl), citric acid (CA), and ammonium oxalate/oxalic acid (AOOA). For the latter two, the conditions used in the literature were followed [9,11]. In all cases, the treatments were carried out in duplicate.

The AIR of three different olive waste (alperujo) samples and the apple, orange, and strawberry fruits were macerated using ChCl at 60 °C for 18 h in order to ensure good contact between the ChCl and the cell wall component (pectin), with the formation of possible DES or at least to form hydrogen bonds that facilitate their extraction. In the second step, pectin extraction from the ChCl samples was carried at 80 °C for 2 h and compared to two commonly used methods for pectin extraction using organic acids of a chelating nature, such as citric acid (CA) and ammonium oxalate/oxalic acid (AOOA) in the same conditions.

#### 2.4.1. Pectin Extraction with Choline Chloride (ChCl)

Approximately 25 g of AIR from the alperujo samples (ALP-1, ALP-2, ALP-3) and 25 g of ChCl were mixed in 250 mL of deionized water and kept in maceration at 60 °C with continuous agitation in a rotatory evaporator, without a vacuum, for 18 h. The AIR samples obtained from fruit (apple, orange, and strawberry) were processed in a similar way. The procedure was followed according to the preparation of natural deep eutectic solvent (NADES) using the heating method described by Dai, van Spronsen, Witkamp, Verpoorte, and Choi. (2013) [23], although with some modifications. In this case, we prolonged the maceration time, with gentle stirring and rotational agitation in a round-bottomed flask, in order to ensure good contact between molecules. Subsequently, the mixture was treated at 80 °C for 2 h in a shaking water bath. After the extraction solutions were centrifuged at 3000× *g* for 15 min, the liquids obtained were precipitated with ethanol to a final concentration of 80%, and the mixture was kept at 4 °C overnight. The samples were then centrifuged at 3000× *g* for 10 min. The precipitated pectins were redissolved in deionized water and subsequently dialyzed for removing phenols, pigments, monosaccharides, and other low-molecular-mass components. This dialysis was carried out against deionized water for 48 h, using dialysis bags of cellulose with a molecular cut-off of 12,000 Da, with water changed every 12 h. Finally, the pectins were lyophilized at a reduced pressure of 0.3 mbar and at −50 °C. The yield was determined gravimetrically and expressed as the ratio of pectin mass obtained to the initial mass of AIR.

#### 2.4.2. Pectin Extraction Using Citric Acid (CA) and Ammonium Oxalate/Oxalic Acid (AOOA)

Approximately, 25 g of AIR from the alperujo samples (ALP-1, ALP-2, ALP-3) was added to 1 L of deionized water, corresponding to the ratio 1:40 (AIR: water), and then the pH was adjusted to 2.2 using 1.9 g of citric acid as the extracting agent or the pH adjusted to 3.4 via the addition of 16 g of ammonium oxalate and 4 g of oxalic acid as extracting agents. The same procedure was used for the AIR obtained from the fruit samples (apple, orange, and strawberry). The extraction process was carried out at 80 °C for 2 h in a shaking water bath. The supernatants were obtained using centrifugation and the precipitated pectins as previously described in Section 2.4.1.

### 2.5. Chemical Characterization of Pectin Extracts

An estimation of the galacturonic acid (Gal A) content was obtained via hydrolysis of the pectin with concentrated sulfuric acid, followed by quantification of the Gal A in a standard solution according to the spectrophotometric method described by Blumenkrantz and Asboe-Han for uronic acids [24]. It is worth noting the very high proportion of Gal A with respect to the glucuronic acid found in alperujo samples, which has been described in previous work [25]. Also, for the other pectins from the fruit samples, which are composed mainly of galacturonic polymers, analysis of the uronic acid was the method to estimate the galacturonic acid present in the samples. To determine the individual neutral sugar composition, the pectin samples were hydrolyzed using 2 N trifluoroacetic acid (TFA) at 121 °C for 1 h and then further converted into alditol acetates (reduction and acetylation). The resulting alditol acetates were analyzed using gas chromatography (GC) [26], inositol as internal standard, and an HP-5890 series II gas chromatograph fitted with a 30 m × 0.25 mm fused silica capillary column (SP-2330 from Supelco, Bellefonte, PA, USA). The chromatographic conditions of GC were described by Lama-Muñoz, Rodríguez-Gutiérrez, Rubio-Senent, and Fernández-Bolaños, (2012) [25]. The molar constituent composition of each pectin extract was used to calculate the proportion of the different pectin structural domains. The pectin backbone is composed of HG (100% Gal A) and RG I ((Rha): GalA of 1:1). Therefore, the relative molar ratio of Rha and GalA inferred that HG = GalA − Rha and RGI = GalA − HG + Rha [10].

The total sugar content was determined as the sum of rhamnose (Rha), fucose (fuc), arabinose (ara), xylose (xyl), mannose (man), galactose (gal), and glucose (glc) on a weight basis. The total phenolic content was estimated using the Folin–Ciocalteu spectrophotometric method and expressed as grams of gallic acid equivalents [27]. The total protein content was determined using the Bradford method [28], using the Coomassie Brilliant Blue G-250 reagent. Absorbance was measured at 595 nm. Bovine serum albumin was used as the standard. All the colorimetric measurements were carried out using the iMark™ Microplate Absorbance Reader (Bio-Rad Laboratories, Madrid, Spain).

The degree of methylation (DM) was measured via the estimated methanol released by the saponification of pectins with 2M NaOH at 20 °C for 60 min [29]. The amount of methanol released was measured enzymatically using alcohol oxidase (0.01 unit/μL in 100 mmol/L citrate buffer, pH = 6.5) and a spectrophotometric method, as described previously by Galanakis, Tornberg, and Gekas, (2010) [30], using a standard solution containing methanol for calibration. The DM was calculated as the moles of methanol measured per mole of uronic acid.

The molecular characteristics were determined using high-performance size-exclusion chromatography (HPSEC), using Jasco equipment (LC Net II ADC, Kyoto, Japan) equipped with a refractive index detector (Jasco RI-1530) and a TSKgel GMPWXL (dextran MW < 50,000 KDa) column (300 × 7.8 mm i.d., Tosoh Bioscience LLC, King of Prussia, PA, USA). The samples were eluted using distilled water at a flow rate of 0.5 mL/min containing 0.01M trifluoroacetic acid. The system was calibrated with dextrans of 252, 110, 70, and 40, 6 kDa, and glucose.

### 2.6. Spectroscopic Characterization of Pectin Extracts

The infrared spectroscopic spectra (FT-IR) of the pectin were determined using a Bruker Invenio spectrophotometer using a platinum attenuated total reflectance (ATR) accessory, equipped with a diamond crystal (Bruker Optics GmbH & Co.KG, Ettllingen, Germany). The spectra were acquired by scanning from the frequency range of 400–4000 cm^−1^, with a resolution of 4 cm^−^^1^ resolution and an average of 16 scans per sample.

1H NMR spectroscopy was performed at 25 °C and pH 4.5–4.8 for the ChCl pectin extract, using a Bruker Avance NEO 500 spectrometer equipped with a BBFO 5 mm dual resonance Z-gradient probe and using D2O as the solvent. Chemical shifts were expressed in δ units (ppm) relative to the solvent peak (D2O) assigned at 4.79 ppm. The conditions used for the experiments were 500 MHz; pulse power 90°; relaxation delay 10 s; number of scans 32; number of dummy scans 4; and acquisition time (AQ) 3.41 s.

### 2.7. Statistical Analysis

The STATGRAPHICS^®^ plus 4.0 software was used for statistical analysis. The results were expressed as mean values of analytical triplicates ± standard deviation. Analysis of variance (ANOVA) as well as the least significant difference (LSD) method were applied to compare the means. Results were statistically significant at a *p*-value < 0.05 calculated at a confidence level of 95%.

## 3. Results and Discussion

### 3.1. Extraction Yields and Chemical Characterization of Pectin Extracted Using Choline Chloride (ChCl)

Table 1a and Table 1b show the yield and chemical composition of the extracted pectins. The yields on AIR dry weight basis of the three alperujo pectins obtained using ChCl ranged from 2.2 to 2.9. kg/100 kg, and showed a 30 to 70% higher yield than using CA. However, the highest yield found (3.9–5.4 kg/100 kg) was obtained using AOOA, which is 45 to 60% higher than ChCl extraction. The efficiency of chelating agents for pectin extractions depends on the Ca^2+^ content and the distribution of free acid groups in the HG chain [7]. The extraction efficiency in the case of apple, orange, and strawberry pectin was different among them and with respect to olive waste. In these three cases, the percentages of pectin obtained using CA were the highest, and exceeded 40% for apple and strawberry and almost 70% for orange with respect to ChCl extraction. The strong acid condition improves the release and dissolution of pectin from the cell wall [31]. However, the yield of pectin extracted from apples using ChCl was even slightly higher than that of the AOOA agent (*p* < 0.05). These results indicate that ChCl interacts with the cell wall, which is a priori a good pectin extracting agent and is influenced by the cell wall of the plant source.

The chemical composition of the extract obtained using organic acids (CA and AOOA) from various sources together with that of ChCl for comparison are also shown in Table 1a and Table 1b. The main components in all pectin samples were the uronic acid (UA) and neutral sugar (NS) contents, and, to a lesser extent, the protein and phenolic compounds. The value of UA in ChCl extraction ranged from 45.9 to 53.5 kg/100 kg (dry basis) (S.I.) for the olive waste pectin, and 70.6 kg/100 kg for apple, 61.7% for orange, and 68.6 kg/100 kg for strawberry pectins, which showed the highest proportion compared to those found by the two commonly used agents for pectin extraction. Statistical analysis of these UA values indicated that ChCl has a similar or significant effect (*p* < 0.05) with respect to the other extraction agents. Their high contents in UA and lower concentration of different organic compounds indicate the great purity of the pectin extracted using ChCl. In addition, although in some cases, the yields of the ChCl pectin decreased, the pectin quality was in accordance with the lower content of NS, which occurred in alperujo pectin as well as in the strawberry pectin, where, in this last case, the contents of phenolic compounds and proteins were also lower. The contents of UA and NS were very similar using the three extracting agents in the case of orange pectin, while the content of UA and NS in apple pectin extracted using ChCl was highest in comparison to the other organic acid agents. Therefore, the results show that this greater or lesser purity of the pectin not only depends on the type of starting material but is also influenced by the extraction agent employed [7,32].

Although there are differences among the compositions of NS of all the pectic material, they all have arabinose as the main neutral sugar residue, except for strawberry pectin, with the galactose at a similar level or slightly higher than arabinose. In addition, arabinose together with other NS including rhamnose, fucose, xylose, and galactose, which comprise the branched region of pectin, showed a much lower content in the pectin obtained using ChCl than the one obtained using CA, except for orange pectin, which was very similar in both cases. Likewise, except for orange pectin, in all the other samples, the pectin obtained using ChCl contained a higher homogalacturonan (HG) proportion than the pectins obtained using CA and AOOA. Pectin with a higher proportion of arabinose and galactose was obtained from olive waste (alperujo) rather than from fruits, which could be part of the RG-I as arabinan, galactan, or arabinogalactan chains [33,34]. In addition, the presence of glucose and mannose in the alperujo pectins (ALP-1, ALP-2 and ALP-3) obtained using CA was higher than in the ones extracted using AOOA and ChCl, possibly indicating the extraction of non-pectic polysaccharides such as hemicelluloses together with the pectin molecule, or the presence of RG-II, although this RG-II has not been described in olive pomace pectin [33].

Taking into account the molar ratio of UA to NS, as a measure of the linearity of pectin [35], and the molar ratio of the sum of arabinose and galactose to UA as a measure of the degree of branching [36] for all samples, except orange pectin, the ChCl pectin showed much more linearity and was less branched than the pectin obtained using CA, and more closely related to the one obtained using AOOA, such as apple pectin. The strawberry pectin extracted using ChCl was the most linear with the fewest neutral sugar side chains, showing ratios of 13.7 for linearity and 0.05 for branching, which means that it is mainly a HG region. Whereas the pectin with the lowest linearity was the one extracted using CA from ALP-1, presenting a pectin ratio of 0.89 and the highest degree of branching with a ratio of 1.03, its linearity increased using ChCl to up to 1.92, and the degree of branching decreased to 0.47. Interestingly, these results are contrary to those found previously when the alperujo was treated directly using ChCl under the same conditions of this study, with lower UA content, a short backbone of the HG domain, and a higher proportion of RG-I than the one treated with CA [20]. ChCl, as a good proton acceptor, seems to act differently within the alperujo by forming hydrogen bonds with compounds which are not present in the isolated cell wall. In both cases, the mechanism of the intermolecular interaction between ChCl and certain compounds present in the cell wall or alperujo matrix will have to be investigated.

Furthermore, as shown in Table 1a and Table 1b, all isolated pectin had a moderate to high degree of methylation (DM), which ranged from 33.7% for alperujo pectin to 87.1% for apple pectin. However, newly isolated pectin using ChCl for all samples showed a DM which was significantly lower than that recovered from all sources using CA and AOOA. Although the mechanism by which the ChCl acts is unknown, the results suggest an effective solubilization of the highly demethylated pectin region, possibly via the hydrogen bond between the free carboxyl group of the pectin molecule and the ChCl, which facilitates the disruption of the cell wall and its extraction. Since the initial pH in the extraction of the pectins using ChCl ranged from 4.5 to 4.8, possible demethylation by high pH was not discounted [37].

The molecular-sized distribution of the pectin extracts obtained for each extracting agent as determined using high-performance size exclusion chromatography (HPSEC) is shown in Figure 1. Due to different molecular shapes or hydrodynamic volumes and the density of the pectins, it is difficult to ascertain their molecular mass using HPSEC without using a specific detection such as light scattering or on-line viscosity [38]. However, a refractive index (RI) detector was used for rapid characterization of the pectin molecular size (MS). The RI profiles of the three samples of alperujo pectin obtained using the three extracting agents tested (Figure 1) appear quite similar and they are composed of 3–4 recognized zones of macromolecule populations, by which only the relative amounts of each zone varied. However, major differences in a greater proportion of smaller molecular sizes could be observed when the different alperujo samples were compared among themselves, possibly being influenced by the action of the enzyme during the different storage times of the olive waste or different degrees of ripening of the olive fruits [39]. The elution patterns showed a high proportion of high MS material at retention times of 11–12 min, which was only degraded in the ALP-3 sample pectin (Figure 1) obtained using ChCl, with the formation of fragments with lower MS between 13.5–15.5 min. In addition, a shoulder-shaped zone can be recognized at 15.5 min, which is in the samples obtained using AOOA, and much softer. It was followed by a zone with the lowest MS at 19–21.5 min or even at 22.5–23.5 min, which was the pectin fragment obtained using AOOA, which was somewhat more resistant to degradation, while the fragments obtained using CA or ChCl were similar.

From the overlay chromatogram of the elution of the three fruit pectins using the three extracting agents shown in Figure 1, it can be concluded that the pectin profiles which depended on the extracting agents exhibited minor differences, while those which depended on the type of starting material varied. The analysis revealed that the three samples showed a high proportion of high MS with a broader distribution, with a retention time that varied between 10 and 15 min for apple and orange pectin (Figure 1), and between 9 and 13 min for strawberry (Figure 1). The ChCl pectin extracted from apples and oranges showed similar behavior with a shoulder at 13–13.5 min, which was not evident in the CA or AOOA samples, although this effect did not occur in the strawberry pectin. Next, small differences could be observed in fragments with lower MS, with varying relative amounts for the pectin from apple and orange extracted using the three extracting agents, with a high proportion of fragments at the retention time of 23.5 min for the apple pectin extracted using CA, probably due to acid hydrolysis. In the strawberry pectin (Figure 1), the profiles of the ChCl and CA pectin were almost identical, with three populations, a high proportion of high MS followed by a second peak at 15.5 min of retention time and also a high proportion of a third peak at 21.5 min.

### 3.2. Spectroscopy Analysis of the Different Pectin Preparations

#### 3.2.1. FT-IR Analysis

The FT-IR spectra of the isolated pectin samples (Figure 2) with the three extracting agents showed that the main structure of pectin was not greatly affected by the ChCl compared to the extraction of the two-acid chelator. The main functional groups of pectin were found in the spectra of the ChCl pectin as well as for the CA and AOOA pectin from all plant sources, with only slight differences in the intensity of some bands. The spectra showed a characteristic absorption band at 3400 cm^−1^ attributed to the OH-stretching vibration of the hydroxyl group of polysaccharides, and peaks at 2927 cm^−1^ and 2850 cm^−1^, not well defined in many cases, which correspond to the asymmetric and symmetric CH stretching of the aliphatic carbon of galacturonic acid [40]. In all the samples except for the alperujo pectin extracted using CA, a doublet band appeared at 2360 cm^−1^ and 2340 cm^−1^, which can be attributed to the aliphatic C–H bonds of the methyl ester group (−OCH3) [41]. The ranges of the absorption from 850 cm^−1^ to 1300 cm^−1^, known as the fingerprint region of the pectin, using which different types of pectin can be distinguished, were almost identical between the ChCl pectin and the other pectins from each source and were attributed to vibration in the C–O–C and C–OH of the monosaccharides and the presence of glycosidic bonds from polysaccharide chains.

The main difference found in all the samples was the ratio between the bands at about 1740 cm^−1^ and 1630 cm^−1^, which are characteristics of the C=O stretching vibration of the esterified carbonyl (COO-R) group and non-esterified carboxylic (COO^−^) group, respectively, which come from the UA in pectin [42]. For all the ChCl pectins, the absorption bands at 1630 cm^−1^ increased, up to 1740 cm^−1^ for the pectin isolated using CA and AOOA. In other words, the intensity of the peak ratio of the free carboxyl to the ester carbonyl group was higher for all ChCl pectins, which indicates that these ChCl pectins are lower in methoxyl. This finding is in agreement with the chemical analysis above, where the results of the DM of ChCl pectin were always lower than those of the other pectins. Since the initial pH in the ChCl extraction ranged from 4.5 to 4.8, some chemical demethoxylation of pectin can occur as this occurs at a pH close to neutral (4–6) [37]. Therefore, the results suggest, as mentioned above, that ChCl can form hydrogen bonds with carboxylic acid by accepting or donating protons, facilitating the extraction of the demethoxyl pectin region. However, the possibility of electrostatic bonds between the choline ions and the carboxylic groups of the pectin also exists.

#### 3.2.2. ^1^H NMR Analysis

^1^H NMR spectroscopy analysis was further performed to better understand the structural information of the isolated pectin samples using the three extracting agents (Figure 3). All the signals found are consistent with the previously reported data on pectin. The spectrum of each sample contains a sharp signal at 3.74 ppm, which corresponds to protons in the methyl ester group binding to the carbonyl group of galacturonic acid [43,44]. Also, in all the samples, with a slight difference in the intensity of signals, weak chemical shifts at 2.00 ppm and 2.10 ppm derived from the acetyl groups binding at 2-O- and 3-O- acetylated galacturonic acid were clearly observed [45]. This confirmed that the structure of ChCl pectin does not differ from the CA and AOOA pectins with predominantly esterified galacturonic acid in all samples. Other major signals were assigned to the five protons in D-galacturonic acid: H-1, 5.03 ppm; H-2, 3.67 ppm; H-3, 3.88 ppm; H-4, 4.21 ppm; H-5, 4.88 ppm. The signals at 1.18 ppm and 1.23 ppm were attributed to the methoxyl group of O-2- and O-2,4-linked rhamnose, respectively [46,47]. In the anomeric region, the signals at 5.07 ppm and 5.17 ppm were attributed to the H-1 of arabinose and rhamnose, respectively. The arabinose H-4 signal at 4.07 ppm and rhamnose H-4 signal at 3.28 ppm were also present in all the samples, which is in agreement with previous reports [48]. These findings indicate the presence of the backbone of RG-I, with rhamnose bound to galacturonic acid and with some side chains of other neutral sugars attached.

The ^1^H NMR spectrum of the ChCl pectin shows the presence of signals which are within the reported region of chemical shifts in the literature for choline [14]. The ChCl remains associated with the pectin after extraction and an extensive dialysis, probably due to hydrogen interactions since ChCl is a good hydrogen bond acceptor. Likewise, the spectra of the CA pectin samples shows a group of doublets of doublet signals at shifts from 2.47 to 2.61 ppm, which correspond to the methylene protons from citric acid moieties [49]. The residual amount of CA in the extracted orange pectin was high and remained associated with the pectin. This is in agreement with the report by Kermani et al. (2014) [6], who showed that it is difficult to fully remove CA from pectin after extraction, probably due to some ionic interaction. In addition, in the strawberry and apple pectins, a doublet of doublets of doublets from 3.48 to 3.60 ppm and a doublet from 3.70 to 3.75 ppm appear, which are more abundant in the CA pectins in comparison with the ChCl and AOOA pectins, which could find its explanation by taking into account the stability of the unknown compound with CA during the isolation of pectin, or the residual presence of glycerol from the dialysis membrane with doublets close to those of glycerol. Although the presence of other compounds in the AIR material is probably considerably reduced, the possibility of interaction between the pectin and some another compound should not be forgotten.

The results showed that the effect of ChCl, a quaternary ammonium salt and a good hydrogen bond acceptor, on the pectin yields varied depending on the cell wall plant source compared to the two organic acids with chelating properties (CA and AOOA), currently used for pectin extraction as replacements for mineral acids. The main limitations of pectin extraction using ChCl were the long extraction times and the percentage of ClCh used. However, the use of this innovative technology will be suitable for industrial use at affordable costs when the optimization of the process improves the extraction yield, reducing the time and ChCl amount. It is a green interesting process since it uses a raw material that is renewable and safe for the environment, operator, and consumer. In addition, since the market for ChCl is currently growing for animal and human food, due to its important added health properties [50], this system would generate co-products, instead of waste.

The alperujo pectin extracted using ChCl had a higher yield than CA, but lower than AOOA. For apples, oranges, and strawberry fruits, the pectin yield from CA extraction was the highest, although in the case of apples, ChCl gave a similar pectin yield to that of AOOA. On the other hand, the purity of the pectin extracted using ChCl, which is estimated in terms of the percentage of galacturonic acid, is higher, except for orange pectin, than that of the other extracting agents. Also, in this case, the neutral sugar chains in the ChCl pectin had a lower degree of branching, and were more linear and richer in the homogalacturonan (HG) region. Therefore, since organic acids produce comparatively higher yields and pectin of better quality than mineral acids, although the latter are actually used, i.e., the HNO_3_ industry standard [10], this acid-free pectin extraction process using ChCl seems to be a promising way forward.

This work does not contain information on the functionality of the extracts, such as their viscosity and gel-forming properties. However, HPSEC chromatography together with FT-IR and ^1^H NMR spectroscopy analysis of the pectin extracted using ChCl in comparison with the pectin extracted using CA and AOOA showed that the pectins were quite similar, with a lower degree of methylation and with a certain residual amount of choline partially associated with the extracted ChCl pectin. The influence of these two latter parameters on its functional properties should be studied in future research.

## 4. Conclusions

In this study, an approach to the use of ChCl was investigated for the extraction of pectin from different complex cell wall matrices. These findings demonstrated that ChCl has the capability to act as an effective medium for the extraction of both dissociated and undissociated forms of acidic components from cell wall networks such as pectin. The solubilization of pectins as an important part of the wall material demonstrates that the use in this case of a component widely used in the formation of DES, such as choline chloride, can help at an industrial level in the valorization of agri-food by-products through the extraction of all their bioactive components. Further studies will be necessary with each type of complete by-product, analyzing the solubilization of its bioactive components.

Depending on the plant source and the functionality of the isolated pectin using ChCl, it could be considered as a potential extracting agent and an acid-free alternative in the process of pectin production, whose mechanism of pectin extraction from the cell wall should be investigated.

## Figures and Tables

**Figure 1 foods-12-04166-f001:**
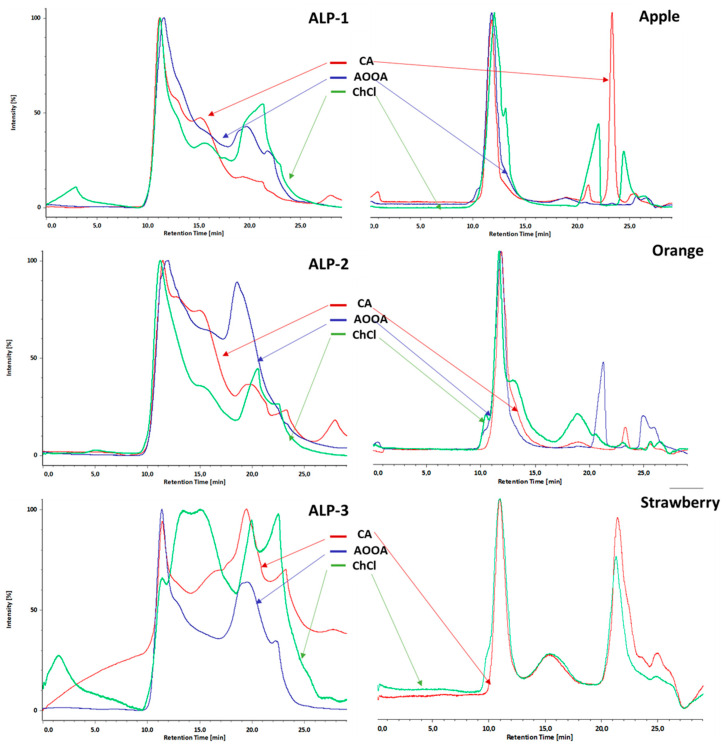
High-performance size-exclusion chromatography coupled with refractive index detector chromatographic profile of ChCl, AOOA, and CA pectins obtained from three different alperujo samples (ALP-1, ALP-2, ALP-3) and apples, oranges, and strawberries.

**Figure 2 foods-12-04166-f002:**
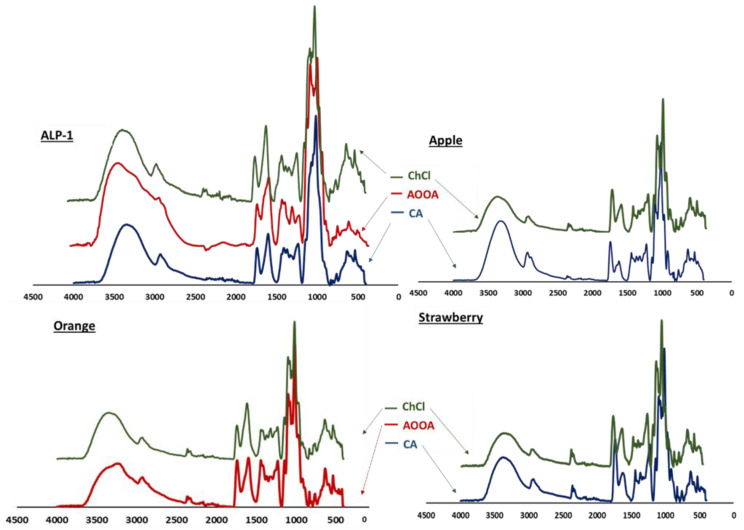
FT-IR spectra of ChCl, AOOA, and CA pectins obtained from one alperujo (ALP-1) and apple, orange, and strawberry samples.

**Figure 3 foods-12-04166-f003:**
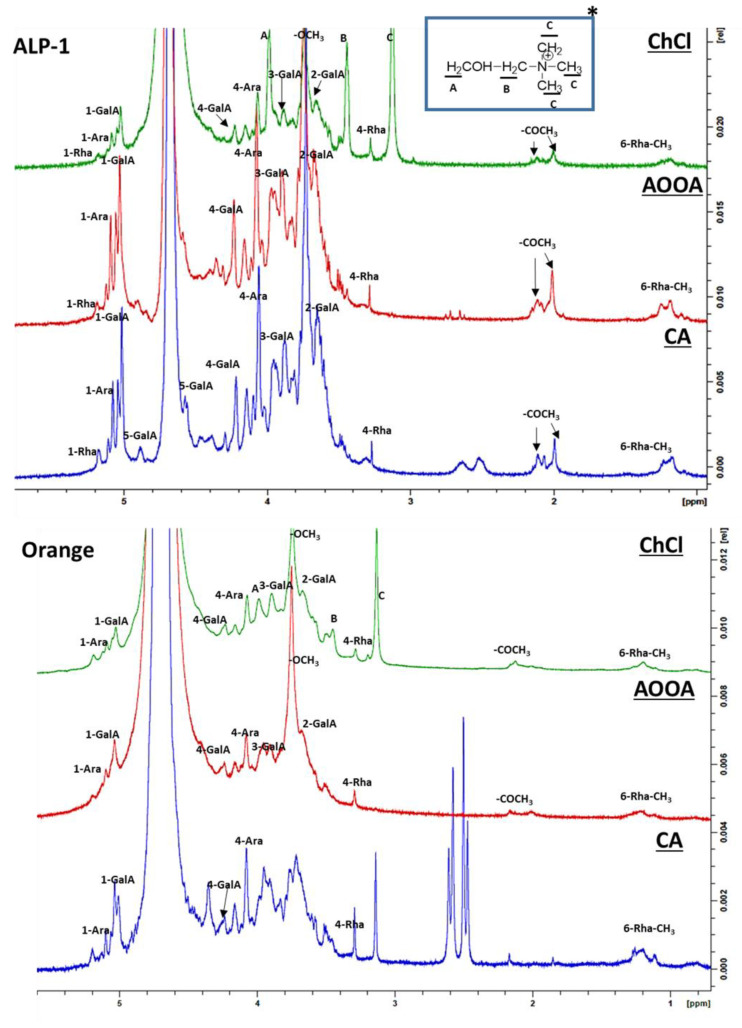
1H-NMR spectra of ChCl, AOOA, and CA pectins obtained from one alperujo (ALP-1) and apple, orange, and strawberry samples. * Choline formula with the carbons A, B and C labeled.

**Table 1 foods-12-04166-t001:** (**a**) Yield (kg/100 kg AIR), chemical composition (kg/100 kg extract), and neutral monosaccharide and uronic acid (UA) compositions (% molar ratio) of the ChCl, AOOA, and CA pectins obtained from three different alperujo samples (ALP-1, ALP-2, ALP-3) and apples, oranges, and strawberries. (**b**) Yield (kg/100 kg AIR), chemical composition (kg/100 kg extract), and neutral monosaccharide and uronic acid (UA) compositions (% molar ratio) of the ChCl, AOOA, and CA pectins obtained from apples, oranges, and strawberries.

(a)									
	ALP-1			ALP-2			ALP-3		
	CA	AOOA	ChCl	CA	AOOA	ChCl	CA	AOOA	ChCl
**Yield**	1.22 ± 0.01 a	3.92 ± 0.07 b	2.20 ± 0.08 c	0.95 ± 0.001 a	5.24 ± 0.21 c	2.88 ± 0.39 b	0.65 ± 0.01 a	5.42 ± 0.18 c	2.28 ± 0.09 b
**Uronic Acid**	30.6 ± 6.41 a	40.8 ± 5.71 b	45.9 ± 6.32 b	39.3 ± 6.01 a	47.7 ± 5.18 b	53.5 ± 3.34 c	34.6 ± 1.48 a	47.5 ± 1.47 b	49.5 ± 9.08 b
**Phenols**	2.21 ± 0.16 a	1.76 ± 0.26 a	2.64 ± 0.22 a	2.84 ± 0.66 b	2.03 ± 0.13 b	1.659 ± 0.15 a	4.89 ± 1.16 b	3.54 ± 0.16 a	6.46 ± 0.95 b
**Neutral Sugar**	37.13 ± 0.91 b	31.84 ± 1.39 a	29.89 ± 5.06 a	28.69 ± 1.5 b	26.66 ± 1.63 b	17.13 ± 0.65 a	30.4 ± 1.70 c	26.32 ± 0.70 b	23.56 ± 0.53 a
**Protein**	0.31 ± 0.005 a	1.21 ± 0.022 b	1.56 ± 0.013b	1.33 ± 0.03 c	0.75 ± 0.015 b	0.31 ± 0.00 a	1.01 ± 0.01 a	3.59 ± 0.03 b	0.79 ± 0.025 a
**Ash**	7.01 ± 0.02 b	5.47 ± 0.08 a	4.3 ± 0.03 a	5.58 ± 0.07 b	3.56 ± 0.02 a	5.05 ± 0.02 b	-	-	-
**Total**	77.2	81.1	83.8	77.7	80.7	77.6	70.9	80.9	80.3
**Total Carbohydrates**	67.7	72.6	75.3	68.0	74.4	70.6	65.0	73.8	73.1
**Rhamnose (Rha)**	1.68	1.71	1.85	1.98	1.86	1.40	3.11	2.78	2.58
**Fucose (Fuc)**	N.D	N.D	0.03	0.03	N.D	N.D	N.D	N.D	N.D
**Arabinose (Ara)**	29.9	28.0	18.5	18.1	22.5	13.8	23.6	22.7	21.3
**Xylose (Xyl)**	2.53	1.74	5.86	2.52	1.56	1.16	2.12	1.91	1.91
**Mannose (Man)**	1.73	1.20	1.47	1.98	1.02	0.94	1.58	1.29	1.08
**Galactose (Gal)**	16.4	9.78	9.95	13.3	7.46	5.89	13.9	13.6	7.85
**Glucose (Glc)**	2.59	1.37	2.03	4.20	1.50	1.41	2.54	2.42	1.29
**Galacturonic acid (GalA)**	45.1	56.2	60.3	57.8	64.1	75.7	53.2	64.3	67.7
**DM ***	71.2	42.4	40.9	85	67.2	50.85	59.8	43.75	33.7
**HG (GalA-Rha)**	43.4	54.4	58.4	55.8	62.3	74.3	50.1	61.6	65.2
**RG I (GalA-HG + Rha)**	3.36	3.42	3.7	3.96	3.72	2.8	6.22	5.56	5.16
**Linearity (GalA/Rha + Fuc + Ara + Xyl + Gal)**	0.89	1.36	1.66	1.61	1.92	3.40	1.25	1.57	2.01
**Branching (Rha + Gal/GalA)**	1.03	0.67	0.47	0.54	0.47	0.26	0.70	0.56	0.43
**(b)**									
	**Apple**			**Orange**			**Strawberry**		
	**CA**	**AOOA**	**ChCl**	**CA**	**AOOA**	**ChCl**	**CA**	**ChCl**	
**Yield**	11.4 ± 0.4 c	6.63 ± 0.05 b	5.36 ± 0.1 a	8.81 ± 0.12 c	5.35 ± 0.01 b	2.82 ± 0.21 a	16.0 ± 0.95 b	9.1 5 ± 0.39 a	
**Uronic Acid**	42.5 ± 5.5 a	52.0 ± 6.0b	70.6 ± 7.7 c	64.5 ± 5.6 b	65.2 ± 8.3 b	61.7 ± 5.4 a	63.3 ± 3.04 a	68.6 ± 2.1 b	
**Phenols**	N.D	N.D	N.D	N.D	N.D	N.D	9.16 ± 0.029 b	6.26 ± 0.039 a	
**Neutral Sugar**	14.2 ± 1.16 b	11.1 ± 0.50 a	16.4 ± 0.42 c	19.9 ± 0.98 b	16.4 ± 0.90 a	23.2 ± 0.18 c	7.56 ± 0.21 a	6.84 ± 1.58 a	
**Protein**	1.04 ± 0.15 a	1.23 ± 0.10 a	1.74 ± 0.03 b	0.94 ± 0.04 a	2.24 ± 0.23 b	2.20 ± 0.43 b	3.66 ± 0.44 b	0.69 ± 0.21 a	
**Ash**	1.47 ± 0.02 b	1.02 ± 0.01 a	3.21 ± 0.14 c	7.68 ± 0.28 b	1.05 ± 0.02 a	7.1 ± 0.32 b	0.55 ± 0.02 a	1.32 ± 0.01 b	
**Total**	59.2	65.4	89.1	93.0	84.9	94.2	84.2	83.6	
**Total Carbohydrates**	56.7	63.1	84.1	84.3	81.6	84.9	86.5	75.3	
**Rhamnose (Rha)**	0.90	0.78	0.74	0.73	0.73	0.83	0.73	0.61	
**Fucose (Fuc)**	0.45	0.43	0.49	0.35	0.25	0.43	0.22	0.28	
**Arabinose (Ara)**	15.30	10.4	12.3	14.1	11.7	14.7	3.1	2.2	
**Xylose (Xyl)**	2.32	1.51	1.45	0.91	0.53	0.93	1.03	1.58	
**Mannose (Man)**	0.40	0.50	0.31	0.78	0.48	1.63	0.44	1.66	
**Galactose (Gal)**	4.96	3.46	3.48	6.1	6.01	7.59	4.4	1.93	
**Glucose (Glc)**	0.72	0.59	0.74	0.60	0.43	1.19	0.76	0.79	
**Galacturonic acid (GalA)**	74.9	82.4	80.5	76.4	79.9	72.7	89.3	90.9	
**DM ***	87.1	72.2	68.5		52.4	39.9	73.3	59.7	
**HG (GalA-Rha)**	74.0	81.6	79.8	75.7	79.1	71.9	88.6	90.3	
**RG I (GalA-HG + Rha)**	1.8	1.56	1.48	1.46	1.46	1.66	1.46	1.22	
**Linearity (GalA/Rha + Fuc + Ara + Xyl + Gal)**	3.13	4.98	4.36	3.44	4.15	2.97	9.42	13.70	
**Branching (Rha + Gal/GalA)**	0.27	0.17	0.20	0.26	0.22	0.31	0.08	0.05	

* Degree of methylation (DM) was expressed in a mol MeOH/mol GalA. The data shown are mean ± standard deviation. Different letters indicate significant differences between each of extracting agents by each alperujo sample and by each fruit sample, while similar letters indicate no significant difference at a *p*-value < 0.05 calculated at a confidence level of 95%.

## Data Availability

The data are contained within the article.

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
