# Peer review of "An Acid-Free Alternative to Pectin Production from the Cell Walls of Olive Oil Waste and Different Fruits Using Choline Chloride"

_foods, 2023, doi:10.3390/foods12224166_

Round 1
Reviewer 1 Report
Comments and Suggestions for Authors
I will admit such extraction techniques are not my field of research (more focused of ultrasonics/high hydrostatic pressure). However, results are promising for the acid-free pectin production. I just have two questions. As extraction treatments were carried in duplicate, was it enough to obtain data for statistical analysis? And do you know approximate particle size (or size distribution) before extraction?
Comments on the Quality of English LanguageMinor errors, just run spell checker or similar tool to correct.
Author Response
I will admit such extraction techniques are not my field of research (more focused of ultrasonics/high hydrostatic pressure). However, results are promising for the acid-free pectin production. I just have two questions. As extraction treatments were carried in duplicate, was it enough to obtain data for statistical analysis? And do you know approximate particle size (or size distribution) before extraction?
Response: Dear reviewer, we appreciate your time and your comments on our manuscript.
Thank you very much for your comments. It is true that that were few replicates, but as it was seen previously that the results were repetitive and the extractions were of many samples and very long processes, we decided finally to do it by duplicates.
Once the alcohol-insoluble residue was obtained (directly ground in a domestic blender and finally dried by solvent exchange and kept overnight in an oven at 40 °C), the particle size was homogeneous and not taken into account for the extraction.
Reviewer 2 Report
Comments and Suggestions for Authors
The manuscript entitled: "An acid-free alternative for pectin production from the cell walls of olive oil waste and different fruits using choline chloride" by Alejandra Bermudez-Oria et al, describe the production pf pectin with choline chloride extraction.
However, the manuscript needs major revision. The specific comments below should help the authors to improve the quality of the manuscript.
Specific comments to manuscript number Foods-2694543:
Line 32-38: It is necessary to use more recent references on the structure of pectins. Pectins have also RG-II and xylogalacturonan domains.
Line 161: What is the concentration of acetone?
Line 187-188: Are AIR and fruit sample put together in deionized water? What does the 1:40 ratio correspond to? the quantity of AIR and fruit?
Line 268-269: “Their high contents in UA indicate the great purity of the pectin extracted with ChCl”: What does that mean? quality based on what parameter?
Line 274: the content of what in apple pectin?
Table 1 and 2: how is GalA quantified? Give the protocol in the materials and methods section.
Line 291: how is the proportion of HG determined?
Table 1 and 2: how are HG, RG, linearity and Branching determined? Give references.
Line 297: is it non-pectic polysaccharide or RG-II domain of pectin?
Line 331: Why was a light scattering detector not used for these characterizations?
Give the Mw and the Mn of each fraction for each sample and each extraction.
Line 348: figure 2 or figure 1?
Line 393-401:
Calculate the DM from the FT-IR data and compare them with the DM calculated by the chemical method.
Figure 2: Why the spectra for ALP-2 and ALP-3 are not presented? Are they identical to ALP-1?
Why all des spectra for all treatment were not presented on the figure 2: example: no spectra for AOOA treatment for apple?
Line 408–441:
Why the spectra for ALP-2 and ALP-3 are not presented? Are they identical to ALP-1?
Figure 3: Why the NMR signals of CA residues were present in ALP-1, orange and not in Strawberry?
The presence of the signal at around 3.15 ppm characteristic of MeOH after a demethylation of pectin seems to be present or not on the spectra independently of the treatment used: what factor in the production of pectins would induce this demethylation?
For each treatment, make sure that the dialysis membrane has been rinsed before use and particularly in the treatment with CA. The doublets present are close to those of glycerol used for preserving the membranes during storage and which must be removed before using the membranes.
A clean spectrum would be appreciated.
Author Response
The manuscript entitled: "An acid-free alternative for pectin production from the cell walls of olive oil waste and different fruits using choline chloride" by Alejandra Bermudez-Oria et al, describe the production pf pectin with choline chloride extraction.
However, the manuscript needs major revision. The specific comments below should help the authors to improve the quality of the manuscript.
Response: Dear reviewer, we appreciate your time and your comments on our manuscript. Based on your comments and questions, we have answers them point by point and made some modifications in the revised MS by yellow colors.
.
Line 32-38: It is necessary to use more recent references on the structure of pectins. Pectins have also RG-II and xylogalacturonan domains
Response: Thanks you for your comments. We have added in the text recent reference on RG-II and xylogalacturonan domains.
Line 161: What is the concentration of acetone?
Response: We have utilized pure acetone. It was corrected in the text.
Line 187-188: Are AIR and fruit sample put together in deionized water? What does the 1:40 ratio correspond to? the quantity of AIR and fruit?
Response: Dear reviewer, thank you very much, it is true that the pectin extraction with ChCl or CA and AOOA is not clear. For clarification of process of extraction of pectin from the cell wall (AIR) the text has been modified.
Line 268-269: “Their high contents in UA indicate the great purity of the pectin extracted with ChCl”: What does that mean? quality based on what parameter?
Response: The purity of pectin extracted with ChCl was estimated in terms of the percentage of galacturonic acid, which is a comparative parameter used by some author and in our case the sentence was completed as follow: high contents in UA and lower concentration of different organic compounds indicate the great purity of the pectin extracted with ChCl. Also two new references [7, 32] were added.
Line 274: the content of what in apple pectin?
Response: You are right. The sentence has been improved.
.
Table 1 and 2: how is GalA quantified? Give the protocol in the materials and methods section.
Response: In accordance with the indication of the reviewer, the method for Gal A quantification, which coincide with uronic acid analysis, has been added in the materials and methods section.
Line 291: how is the proportion of HG determined?
Table 1 and 2: how are HG, RG, linearity and Branching determined? Give references.
Response: In in the materials and methods section the formula for HG and RG-I determination and their reference [10] have been added.
The references on linearity and branching are described in the text [35-36] and the formula are given in Table 1.
Line 297: is it non-pectic polysaccharide or RG-II domain of pectin?
Response: We completely agree with the comments of the reviewer and the two possibilities has been added in the text. However, to our knowledge, the presence of RG-II in olive pomace has not been described. A new reference has been added [33].
Line 331 Why was a light scattering detector not used for these characterizations?
Give the Mw and the Mn of each fraction for each sample and each extraction.
Response: The reason is that we do not have a light scattering detector in our laboratory. We appreciate the comment helping us to improve the paper but HPSEC analysis of pectin is not straightforward, with easy formation of aggregates, where nor refractive index, nor dextrans are the preferred for determining the molecular-size distribution of the pectin extracts. These difficulties are indicated, in part, in the discussion of the results. However, we consider that the RI profiles of the pectin extract from ChCl, CA and AOOA from the different material could be used for comparison.
Line 348: figure 2 or figure 1?
Response: Thank for the useful advice. It is a mistake topographic, the figure 1 is correct.
Line 393-401: Calculate the DM from the FT-IR data and compare them with the DM calculated by the chemical method.
Response: Since the aim was to compare pectin extracted with ChCl, always with a lower degree of methylation than those extracted with CA and AOOA, we consider the chemical analysis sufficient for the numerical value.
Figure 2: Why the spectra for ALP-2 and ALP-3 are not presented? Are they identical to ALP-1?
Why all des spectra for all treatment were not presented on the figure 2: example: no spectra for AOOA treatment for apple?
Response: Indeed, the IR spectra of the alperujo samples were identical, with slight variations in the ratio between the bands of the free carboxyl to the ester carbonyl group, which coincide with the little variation in DM.
Really, we don´t know the reason for the IR spectra of AOOA pectin of apple after repeating it several times always came out very bad.
Line 408-441: Why the spectra for ALP-2 and ALP-3 are not presented? Are they identical to ALP-1?
Figure 3: Why the NMR signals of CA residues were present in ALP-1, orange and not in Strawberry?
Response: Indeed, the 1HNMR spectra of the alperujo samples were identical.
In the case of strawberries, the AIR obtained was scarce and after all chemical analyses we did not have enough samples for their 1HNMR analysis.
The presence of the signal at around 3.15 ppm characteristic of MeOH after a demethylation of pectin seems to be present or not on the spectra independently of the treatment used: what factor in the production of pectins would induce this demethylation?
Response: Dear reviewer you are absolutely right. We have added this possibility in the text. Since the initial pH in the ChCl extraction ranged 4.5-4.8, some chemical demethoxylation of pectin can occur at pH near neutrality (4-6).
For each treatment, make sure that the dialysis membrane has been rinsed before use and particularly in the treatment with CA. The doublets present are close to those of glycerol used for preserving the membranes during storage and which must be removed before using the membranes.
A clean spectrum would be appreciated.
Response: Dear reviewer, you are absolutely right, the doublets are present possibly for the reason you have indicated. Although the dialysis membranes were washed, it seems that it was not enough to completely remove the glycerol. We have added this possibility in the text.
We find it difficult to get cleaner spectra and to be able to include all the comparisons.
Reviewer 3 Report
Comments and Suggestions for Authors
The aim of this study was to extract pectin from the cell walls of olive waste (alperujo), apple, orange and strawberry fruits using choline chloride (ChCl) and compare the yield with extractions with citric acid (CA) and ammonium oxalate/oxalic acid (AOOA). Also chemical and structural characterization were also performed.
The study of the possibility of using quaternary ammonium salt ChCl, as an inexpensive, available, non-toxic, low-viscosity agent with good hydrogen bonding capacity makes the paper very interesting with the subject falling in the scope of the journal.
The paper is, in general, well-written and well- organized. However, English needs improvement. Sometimes sentences are too long and I suggest a revision by a mother-English tongue.
Moreover, about Statistics letters of significant difference reported in Tables 1 and 2 should be corrected: increasing letters should be accompanied by increasing data (number) or vice versa.
Comments on the Quality of English LanguageEnglish needs some improvement. It is advisable to make sentences more shorter.
Author Response
The aim of this study was to extract pectin from the cell walls of olive waste (alperujo), apple, orange and strawberry fruits using choline chloride (ChCl) and compare the yield with extractions with citric acid (CA) and ammonium oxalate/oxalic acid (AOOA). Also chemical and structural characterization were also performed.
The study of the possibility of using quaternary ammonium salt ChCl, as an inexpensive, available, non-toxic, low-viscosity agent with good hydrogen bonding capacity makes the paper very interesting with the subject falling in the scope of the journal.
The paper is, in general, well-written and well- organized. However, English needs improvement. Sometimes sentences are too long and I suggest a revision by a mother-English tongue.
Moreover, about Statistics letters of significant difference reported in Tables 1 and 2 should be corrected: increasing letters should be accompanied by increasing data (number) or vice versa.
Response: Dear reviewer, we appreciate your time and your comments on our manuscript.
We have made a special effort to improve the English and the paper has reviewed by an official native reviewer.
Thank you very much for your kind reminding. The Statistics letters of significant difference were corrected by increasing data.
Reviewer 4 Report
Comments and Suggestions for Authors
The authors present an interesting study on using an acid-free alternative to extracting pectin from olive oil residues and fruits. The article is well-discussed, and the results are noteworthy if they can contribute to reducing food waste. However, some issues require attention:
1. The introduction appears redundant in certain areas. Please organize the text. Also, the objective should be rewritten, avoiding details about the methods applied.
2. Could the authors delineate the primary limitations of their study? Is it possible to address and overcome these limitations? Additionally, what future directions do the authors predict in this area of research?
3. Can you elaborate on the significance of your findings for the food industry? Specifically, how might these advancements influence the utilization of this technique in pectin extraction? Consider facts such as economic implications, culinary application, sustainability, and market trends among others.
4. Please improve the discussion section with future trends and the implications of the findings reported in this study.
Author Response
The authors present an interesting study on using an acid-free alternative to extracting pectin from olive oil residues and fruits. The article is well-discussed, and the results are noteworthy if they can contribute to reducing food waste. However, some issues require attention:
Response: Dear reviewer, we appreciate your time and your comments on our manuscript. Based on your comments and questions, we have answers them point by point and made some modifications in the revised MS by yellow colors.
- The introduction appears redundant in certain areas. Please organize the text. Also, the objective should be rewritten, avoiding details about the methods applied.
Response: In accordance with the indications of the reviewer the introduction has been re-organized. The last paragraph has been moved to Materials and method section and the objective has been clarified.
- Could the authors delineate the primary limitations of their study? Is it possible to address and overcome these limitations? Additionally, what future directions do the authors predict in this area of research?
- Can you elaborate on the significance of your findings for the food industry? Specifically, how might these advancements influence the utilization of this technique in pectin extraction? Consider facts such as economic implications, culinary application, sustainability, and market trends among others.
- Please improve the discussion section with future trends and the implications of the findings reported in this study
Response: The limitation of the study, the relevance for the food industry and future directions have been re-organised and reflected in the final part of the discussion and results. Finally, in Conclusions also include the future trends and the implications of the findings reported in this study.
Round 2
Reviewer 2 Report
Comments and Suggestions for Authors
Specific comments to manuscript number Foods-2694543 :
Ligne 164, 186 : RIA?
Ligne 483 : add reference for the health properties
Ligne 493 : add reference forHNO3 industry standard
Author Response
Thank you for the helpful advice to improve our article.
The topographic mistake and the references have been corrected.